# HIERARCHICAL EMPOWERMENT: TOWARDS TRACTABLE EMPOWERMENT-BASED SKILL LEARNING

## ABSTRACT

General purpose agents will require large repertoires of skills. Empowerment—the maximum mutual information between skills and the states—provides a pathway for learning large collections of distinct skills, but mutual information is difficult to optimize. We introduce a new framework, *Hierarchical Empowerment*, that makes computing empowerment more tractable by integrating concepts from Goal-Conditioned Hierarchical Reinforcement Learning. Our framework makes two specific contributions. First, we introduce a new variational lower bound on mutual information that can be used to compute empowerment over short horizons. Second, we introduce a hierarchical architecture for computing empowerment over exponentially longer time scales. We verify the contributions of the framework in a series of simulated robotics tasks. In a popular ant navigation domain, our four level agents are able to learn skills that cover a surface area over two orders of magnitude larger than prior work.

## 1 INTRODUCTION

How to implement general purpose agents that can execute a vast array of skills remains a central challenge in artificial intelligence research. Empowerment (Klyubin et al., 2005; Salge et al., 2014; Jung et al., 2012; Gregor et al., 2017) offers a compelling objective for equipping agents with a large set of skills. As the maximum mutual information (MI) between skills and the states to which they lead, empowerment enables agents to learn large sets of distinct skills that each target a different area of the state space. The major problem with using empowerment for skill learning is that mutual information is difficult to optimize.

Recent work (Mohamed & Rezende, 2015; Gregor et al., 2017; Eysenbach et al., 2019) has made some progress using Reinforcement Learning (RL) (Sutton & Barto, 1998) to optimize a variational lower bound on mutual information, but these methods are limited to learning small spaces of skills. The main issue is that the reward function produced by the conventional MI variational lower bound does not incentivize skills to target particular regions of the state space when there is significant overlap among the skills. A potentially better alternative is to use Goal-Conditioned Reinforcement Learning (GCRL) as the variational lower bound to MI (Choi et al., 2021). The GCRL variational lower bound employs a reward function that always encourages skills to target specific states regardless of the degree to which the skills are overlapping. However, a key problem with GCRL is that it requires a hand-crated goal space. Not only does this require domain expertise, but a hand-crafted goal space may only achieve a loose lower bound on empowerment if the goal space is too small or too large as the agent may learn too few skills or too many redundant skills. In addition, it is unclear whether goal-conditioned RL can be used to compute long-term empowerment (i.e., learn long-horizon skills) given the poor performance of RL in long horizon settings (Andrychowicz et al., 2017; Nair et al., 2018; Trott et al., 2019; Levy et al., 2019; Nachum et al., 2018).

We introduce a framework, *Hierarchical Empowerment*, that takes a step toward tractable long-term empowerment computation. The framework makes two contributions. The first contribution, *Goal-Conditioned Empowerment*, is a new variational lower bound objective on empowerment that improves on GCRL by learning the space of achievable goal states. We show that after applying the reparameterization trick (Kingma & Welling, 2014), our mutual information objective with respect

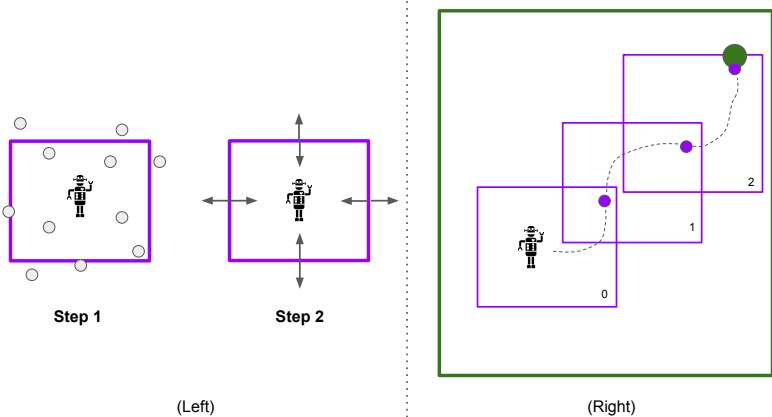

Figure 1: (Left) Illustration of the two step process involved in computing Goal-Conditioned Empowerment. In step one, the goal-conditioned policy is trained with goals ((x,y) locations in this example) sampled from inside and nearby the current goal space (purple rectangle). In step two, the shape and location of the current goal space is updated (e.g., by shrinking, expanding, and/or shifting) to reflect the largest space of goals that can be reliably achieved by the goal-conditioned policy. (Right) Illustration of the hierarchical architecture for an agent that learns two levels of skills. The action space for the top level goal-conditioned policy is set to the goal space of the bottom level goal space policy (purple rectangles). Tasked with achieving a goal (green circle) from the top level goal space (green rectangle), the top level goal-conditioned policy executes a sequence of three subgoals (purple circles), each of which must be contained within the bottom level goal spaces (purple rectangles).

to the learned goal space is a maximum entropy (Ziebart et al., 2008) bandit problem, in which the goal space is rewarded for being large and containing achievable goals. Optimizing the full Goal-Conditioned Empowerment objective takes the form of an automated curriculum of goal-conditioned RL problems. Figure 1(Left) illustrates this idea.

The second contribution of our framework is a hierarchical architecture for scaling Goal-Conditioned Empowerment to longer time horizons. Under this architecture, Goal-Conditioned Empowerment is implemented at multiple levels of hierarchy. What distinguishes each level is the action space for the goal-conditioned policy, which is set to the learned goal space at the level below. This architecture enables each level up the hierarchy to compute empowerment at an exponentially increasing time scale. At the same time, each policy need only learn a short sequence of decisions, making optimizing empowerment with RL more tractable. Figure 1(Right) illustrates this architecture.

Note that our framework has two significant limitations. First, similar to other empowerment-based skill-learning methods, our framework assumes the agent has access to a model of the environment's transition dynamics. Second, the framework assumes the learned goal space is a uniform distribution over a subset of the state space, which significantly limits the types of domains to which the framework can be applied. We discuss these limitations in more detail in section 3.

We evaluate the two proposed contributions of our framework in a series of simulated robotic navigation tasks. The experimental results support both contributions. Goal-Conditioned Empowerment outperforms the baselines at learning the short-term empowerment of states. In addition, our results indicate that hierarchy is useful for computing long-term empowerment as agents with more levels of hierarchy outperformed those with fewer. In our largest domain, our agents are able to complete navigation tasks that are orders of magnitude longer than related work. A video presentation of our results is available at the following url: `https://www.youtube.com/watch?v=OOjtC30VjPQ`.

## 2 BACKGROUND

### 2.1 GOAL-CONDITIONED MARKOV DECISION PROCESSES (MDPs)

A core concept in our work is Goal-Conditioned MDPs (Kaelbling, 1993; Schaul et al., 2015), which represent a decision making setting in which an agent needs to learn a distribution of tasks. We define Goal-Conditioned MDPs by the tuple $\{\mathcal{S}, p(s_0), \mathcal{G}, p(g), \mathcal{A}, T(s_{t+1}|s_t, a_t), r(s_t, g, a_t)\}$. $\mathcal{S}$ is the space of states; $p(s_0)$ is the distribution of starting states; $\mathcal{G}$ is the space of goals and is assumed to be a subset of $\mathcal{S}$; $p(g)$ is the distribution of goals for the agent to achieve; $\mathcal{A}$ is the action space; $T(s_{t+1}|s_t, a_t)$ is the transition dynamics of the environment; $r(s_t, g, a_t)$ is the reward function for a (state,goal,action) tuple. A policy $\pi_\theta(\cdot|s_t, z)$ in a Goal-Conditioned MDP is a mapping from states and goals to a distribution over the action space.

The objective in a goal-conditioned MDP is to learn the parameters $\theta$ of the policy that maximize the sum of discounted rewards averaged over all goals and goal-conditioned trajectories:

$$\arg\max_\theta \mathbb{E}_{g\sim p(g), \tau\sim p(\tau|g)}[\sum_{t=0}^\infty \gamma^t r(s_t, g, a_t)],$$

in which $\tau$ is a trajectory of states and actions $(s_0, a_0, s_1, a_1, \dots)$. The goal-conditioned MDP objective can be optimized with Reinforcement Learning, such as policy gradient methods (Sutton et al., 1999a).

### 2.2 THE SKILL CHANNEL AND EMPOWERMENT

Central to our approach for skill learning is the observation that skills are an instance of noisy channels from Information Theory (Cover & Thomas, 2006). The proposed skill is the message input into the noisy channel. The state that results from executing the skill is the output of the channel. We will refer to this noisy channel as the skill channel. The skill channel is defined by the tuple $\{\mathcal{S}, \mathcal{Z}, p(s_n|s_0, z)\}$, in which $\mathcal{S}$ is the set of states, $\mathcal{Z}$ is the set of skills that can be executed from a start state $s_0 \in \mathcal{S}$, $n$ is the number of actions contained within a skill, and $p(s_n|s_0, z)$ is the probability of a skill $z$ that started in state $s_0$ terminating in state $s_n$ after $n$ actions. That is, $p(s_n|s_0, z) = \int_{a_0, s_1, a_1, \dots, s_{n-1}, a_{n-1}} \pi(a_0|s_0, z) p(s_1|s_0, a_0) \pi(a_1|s_1, z) \dots \pi(a_{n-1}|s_{n-1}, z) p(s_n|s_{n-1}, a_{n-1})$. The distribution $p(s_n|s_0, z)$ thus depends on the skill-conditioned policy $\pi(a_t|s_t, z)$ and the transition dynamics of the environment $p(s_{t+1}|s_t, a_t)$.

The mutual information of the skill channel, $I(Z; S_n|s_0)$, describes the number of distinct skills that can be executed from state $s_0$ given the current skill distribution $p(z|s_0)$ and the skill-conditioned policy $\pi(a_t|s_t, z)$. Mathematically, the mutual information of the skill channel can be defined:
$$I(Z; S_n|s_0) = H(Z|s_0) - H(Z|s_0, S_n).$$
That is, the number of distinct skills that can be executed from state $s_0$ will be larger when (i) the entropy $H(Z|s_0)$ (i.e., the size) of the skill space is larger and/or (ii) the conditional entropy $H(Z|s_0, S_n)$ is smaller (i.e., the skills are more differentiated as they target distinct regions of the state space).

Given our goal of equipping agents with as large a skillset as possible, the purpose of this paper is to develop an algorithm that can learn a distribution over skills $p_\phi(z|s_0)$ and a skill-conditioned policy $\pi_\theta(s_t, z)$ that maximize the mutual information of the skill channel:
$$\arg\max_{p_\phi, \pi_\theta} I(Z; S_n|s_0) \tag{1}$$

The maximum mutual information (equivalently, the channel capacity) of the skill channel is known as Empowerment (Klyubin et al., 2005). Note that in the literature empowerment is often defined with respect to more simple skills, such as single primitive actions or open loop action sequences (Klyubin et al., 2005; Mohamed & Rezende, 2015), but we use the more general definition developed by Gregor et al. (2017) that permits learnable closed loop skills.

### 2.3 CHALLENGES WITH COMPUTING EMPOWERMENT

One major difficulty with calculating empowerment (i.e., maximizing the mutual information of the skill channel) is that computing the posterior distribution $p(z|s_0, s_n)$, located within the conditional

entropy term $H(Z|s_0, S_n)$, is often intractable. Analytically calculating the posterior distribution would require integrating over the intermediate actions and states $a_0, s_1, a_1 \ldots, s_{n-1}, a_{n-1}$. Inspired by Mohamed & Rezende (2015), many empowerment-based skill-learning approaches (Gregor et al., 2017; Eysenbach et al., 2019; Zhang et al., 2021; Strouse et al., 2021; Warde-Farley et al., 2018; Baumli et al., 2020; Achiam et al., 2018; Hansen et al., 2020) attempt to overcome the issue by optimizing a variational lower bound on the mutual information of the skill channel. In this lower bound, a learned variational distribution $q_\psi(z|s_0, s_n)$ parameterized by $\psi$ is used in place of the problematic posterior $p(z|s_0, s_n)$. (See Barber & Agakov (2003) for original proof of the lower bound.) The variational lower bound is accordingly defined

$$I^{VB}(Z; S_n|s_0) = H(Z|s_0) + \mathbb{E}_{z \sim p(z|s_0), s_n \sim p_\theta(s_n|s_0, z)}[\log q_\psi(z|s_0, s_n)], \tag{2}$$

in which $\theta$ represents the parameters of the skill-conditioned policy $\pi_\theta(a_t|s_t, z)$.

The common approach in empowerment-based skill learning is to train the learned variational distribution $q_\psi$ with maximum likelihood learning so that the variational distribution approaches the true posterior. The skill distribution $p(z)$ is typically held fixed so $H(Z|s_0)$ is constant and can be ignored. The last term in the variational lower bound, as observed by (Gregor et al., 2017), is a Reinforcement Learning problem in which the reward for a (state,skill,action) tuple is $r(s_t, z, a_t) = 0$ for the first $n-1$ actions and then $r(s_{n-1}, z, a_{n-1}) = \mathbb{E}_{s_n \sim p(s_n|s_{n-1}, a_n)}[\log q_\psi(z|s_0, s_n))]$ for the final action.

Unfortunately, this reward function does not encourage skills to differentiate and target distinct regions of the state space (i.e., increase mutual information) when the skills overlap and visit similar states. For instance, at the start of training when skills tend to visit the same states in the immediate vicinity of the start state, the reward $q_\psi(z|s_0, s_n)$ will typically be small for most $(s_0, z, s_n)$ tuples. The undifferentiated reward will in turn produce little change in the skill-conditioned policy, which then triggers little change in the learned variational distribution and reward $q_\psi$, making it difficult to maximize mutual information.

## 2.4 GOAL-CONDITIONED RL AS A MUTUAL INFORMATION VARIATIONAL LOWER BOUND

Choi et al. (2021) observed that goal-conditioned RL, in the form of the objective below, is also a variational lower bound on mutual information.

$$\mathbb{E}_{g \sim p(g|s_0)}[\mathbb{E}_{s_1, \ldots, s_n \sim p_\theta(s_1, \ldots, s_n|s_0, g)}[\log q(g|s_0, s_n)]], \tag{3}$$
$$q(g|s_0, s_n) = \mathcal{N}(g; s_n, \sigma_0)$$

That is, $q(\cdot|s_0, s_n)$ is a diagonal Gaussian distribution with mean $s_n$ and a fixed standard deviation $s_0$ set by the designer. The expression in 3 is a goal-conditioned RL objective as there is an outer expectation over a fixed distribution of goal states $p(g|s_0)$ and an inner expectation with respect to the distribution of trajectories produced by a goal state. The reward function is 0 for all time steps except for the last when it is the $\log$ probability of the skill $g$ sampled from a Gaussian distribution centered on the skill-terminating state $s_n$ with fixed standard deviation $\sigma_0$. Like any goal-conditioned RL reward function, this reward encourages goal-conditioned policy $\pi_\theta(s_t, g)$ to target the conditioned goal as the reward is higher when the goal $g$ is near the skill-terminating state $s_n$.

The goal-conditioned objective in Equation 3, which reduces to $\mathbb{E}_{g \sim p(g|s_0), s_n \sim p(s_n|s_0, g)}[\log q(g|s_0, s_n)]$, is also a variational lower bound on the mutual information of the skill channel. In this case, the variational distribution is a fixed variance Gaussian distribution. Unlike the typical variational lower bound in empowerment-based skill learning, goal-conditioned RL uses a reward function that explicitly encourages skills to target specific states, even when there is significant overlap among the skills in regards to the states visited.

One issue with using goal-conditioned RL as a variational lower bound on mutual information is the hand-crafted distribution of skills $p(g|s_0)$. In addition to requiring domain expertise, if this is significantly smaller or larger than the region of states that can be reached in $n$ actions by the goal-conditioned policy, then maximizing the goal-conditioned RL objective would only serve as a loose lower bound on empowerment because the objective would either learn fewer skills than it is capable of or too many redundant skills. Another issue with the objective is that it is not a practical objective for learning lengthy goal-conditioned policies given that RL has consistently struggled at performing credit assignment over long horizons using temporal difference learning (Levy et al., 2019; Nachum et al., 2018; McClinton et al., 2021).

## 3 HIERARCHICAL EMPOWERMENT

We introduce a framework, Hierarchical Empowerment, that makes two improvements to using goal-conditioned RL as a variational lower bound on mutual information and empowerment. The first improvement we make is a new variational lower bound, *Goal-Conditioned Empowerment*, that uses the same fixed variational distribution as Choi et al. (2021), but also learns the distribution of skills. The second improvement is to integrate a hierarchical architecture that makes computing the empowerment over long time horizons more tractable.

### 3.1 GOAL-CONDITIONED EMPOWERMENT

The Goal-Conditioned Empowerment objective is defined

$$\mathcal{E}^{GCE}(s_0, \theta, \phi) = \max_{p_\phi, \pi_\theta} I^{GCE}(Z; S_n | s_0). \tag{4}$$

$s_0$ is the starting state for which empowerment is to be computed. $I^{GCE}(Z; S_n | s_0)$ is a variational lower bound on the mutual information of the skill channel. $\pi_\theta(s_t, z)$ is the goal-conditioned policy parameterized by $\theta$. We will assume $\pi_\theta$ is deterministic. $p_\phi(z | s_0)$ is the learned distribution of continuous goal states (i.e., skills) parameterized by $\phi$. The mutual information variational lower bound within Goal-Conditioned Empowerment is defined

$$I_{\theta,\phi}^{GCE}(Z; S_n | s_0) = H_\phi(Z | s_0) + \mathbb{E}_{z \sim p_\phi(z | s_0), s_n \sim p_\theta(s_n | s_0, z)}[\log q(z | s_0, s_n)]. \tag{5}$$

Next, we will analyze the variational lower bound objective in equation 5 with respect to the goal-space policy and goal-conditioned policy separately. An immediate challenge to optimizing Equation 5 with respect to the goal space parameters $\phi$ is that $\phi$ occurs in the expectation distribution. The reparameterization trick (Kingma & Welling, 2014) provides a solution to this issue by transforming the original expectation into an expectation with respect to exogenous noise. In order to use the reparameterization trick, the distribution of goal states needs to be a location-scale probability distribution (e.g., the uniform or the normal distributions). We will assume $p_\phi(z | s_0)$ is a uniform distribution for the remainder of the paper. Applying the reparameterization trick to the expectation term in Equation 5 and inserting a fixed variance Gaussian variational distribution similar to the one used by (Choi et al., 2021), the variational lower bound objective becomes

$$I_{\theta,\phi}^{GCE}(Z; S_n | s_0) = H_\phi(Z | s_0) + \mathbb{E}_{\epsilon \sim \mathcal{U}^d}[\mathbb{E}_{s_n \sim p(s_n | s_0, z)}[\log \mathcal{N}(h(s_0) + z; s_n, \sigma_0)]], \tag{6}$$

in which $d$ is the dimensionality of the goal space and $z = g(\epsilon, \mu_\phi(s_0))$ is reparameterized to be a function of the exogenous unit uniform random variable $\epsilon$ and $\mu_\phi(s_0)$, which outputs a vector containing the location-scale parameters of the uniform goal-space distribution. Specifically, for each dimension of the goal space, $\mu_\phi(s_0)$ will output (i) the location of the center of the uniform distribution and (ii) the length of the half width of the uniform distribution. Note also that in our implementation, instead of sampling the skill $z$ from the fixed variational distribution (i.e., $\mathcal{N}(z; s_n, \sigma_0)$), we sample the sum $h(s_0) + z$, in which $h(\cdot)$ is the function that maps the state space to the skill (i.e. goal) space. The change to sampling the sum $h(s_0) + z$ will encourage the goal $z$ to reflect the desired change from the starting state $s_0$. The mutual information objective in Equation 6 can be further simplified to

$$I_{\theta,\phi}^{GCE}(Z; S_n | s_0) = H_\phi(Z | s_0) + \mathbb{E}_{\epsilon \sim \mathcal{U}^d}[R(s_0, \epsilon, \mu_\phi(s_0))], \tag{7}$$
$$R(s_0, \epsilon, \mu_\phi(s_0)) = \mathbb{E}_{s_n \sim p(s_n | s_0, \epsilon, \mu_\phi(s_0))}[\log \mathcal{N}(h(s_0) + z; s_n, \sigma_0)].$$

Equation 7 has the same form as a maximum entropy bandit problem. The entropy term $H_\phi(Z | s_0)$ encourages the goal space policy $\mu_\phi$ to output larger goal spaces. The reward function $\mathbb{E}_{\epsilon \sim \mathcal{U}^d}[R(s_0, \epsilon, \mu_\phi(s_0))]$ in the maximum entropy bandit problem encourages the goal space to contain achievable goals. This is true because $R(s_0, \epsilon, \mu_\phi(s_0))$ measures how effective the goal-conditioned policy is at achieving the particular goal $z$, which per the reparameterization trick is $g(\epsilon, \mu_\phi(s_0))$. The outer expectation $\mathbb{E}_{\epsilon \sim \mathcal{U}^d}[\cdot]$ then averages $R(s_0, \epsilon, \mu_\phi(s_0))$ with respect to all goals in the learned goal space. Thus, goal spaces that contain more achievable goals will produce higher rewards.

The goal space policy $\mu_\phi$ can be optimized with Soft Actor-Critic (Haarnoja et al., 2018), in which a critic $R_\omega(s_0, \epsilon, a)$ is used to approximate $R(s_0, \epsilon, a)$ for which $a$ represents actions by the goal

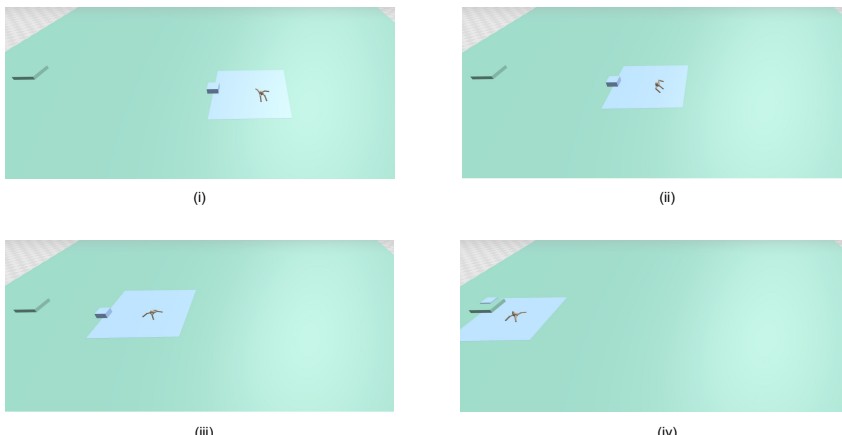

Figure 2: An Ant agent with two levels of skills. The images depict a sample episode in which the top level goal-conditioned policy is tasked with achieving a goal (green box) from the top level goal space (green platform). The top level goal-conditioned policy outputs a sequence of subgoals (blue boxes) that must be contained within the bottom level goal space (blue platforms).

space policy. In our implementation, we fold in the entropy term into the reward, and estimate the expanded reward with a critic. We then use Stochastic Value Gradient (Heess et al., 2015) to optimize. Section F of the Appendix provides the objective for training the goal-space critic. The gradient for the goal-space policy $\mu_\phi$ can be determined by passing the gradient through the critic using the chain rule:

$$\nabla_\phi I^{GCE}(Z; S_n|s_0) = \nabla_\phi \mu_\phi(s_0) \mathbb{E}_{\epsilon \sim \mathcal{U}^d}[\nabla_a R_\omega(s_0, \epsilon, a)|_{a=\mu_\phi(s_0)}]. \tag{8}$$

With respect to the goal-conditioned policy $\pi_\theta$, the mutual information objective reduces to a goal-conditioned RL problem because the entropy term can be ignored. We optimize this objective with respect to the deterministic goal-conditioned policy with Deterministic Policy Gradient (Silver et al., 2014).

In our implementation, we optimize the Goal-Conditioned Empowerment mutual information objective by alternating between updates to the goal-conditioned and goal-space policies—similar to an automated curriculum of goal-conditioned RL problems. In each step of the curriculum, the goal-conditioned policy is trained to reach goals within and nearby the current goal space. The goal space is then updated to reflect the current reach of the goal-conditioned policy. Detailed algorithms for how the goal-conditioned actor-critic and the goal-space actor-critic are updated are provided in section D of the Appendix.

## 3.2 HIERARCHICAL ARCHITECTURE

To scale Goal-Conditioned Empowerment to longer horizons we borrow ideas from Goal-Conditioned Hierarchical Reinforcement Learning (GCHRL), sometimes referred to as Feudal RL (Dayan & Hinton, 1992; Levy et al., 2017; 2019; Nachum et al., 2018; Zhang et al., 2021). GCHRL approaches have shown that temporally extended policies can be learned by nesting multiple short sequence policies. In the Hierarchical Empowerment framework, this approach is implemented as follows. First, the designer implements $k$ levels of Goal-Conditioned Empowerment, in which $k$ is a hyperparameter set by the designer. That is, the agent will learn $k$ goal-conditioned and goal space policies $(\pi_{\theta_0}, \mu_{\phi_0}, \ldots, \pi_{\theta_{k-1}}, \mu_{\phi_{k-1}})$. Second, for all levels above the bottom level, the action space is set to the learned goal space from the level below. (The action space for the bottom level remains the primitive action space.) For instance, for an agent with $k = 2$ levels of skills, the action space for the top level goal-conditioned policy $\pi_{\theta_1}$ at state $s$ will be set to the goal space output by $\mu_{\phi_0}(s)$. Further, the transition function at level $i > 0$, $T_i(s_{t+1}|s_t, a_{i_t})$, fully executes subgoal action $a_i$ proposed by level $i$. For instance, for a $k = 2$ level agent, sampling the transition distribution at the higher level $T_1(s_{t+1}|s_t, a_{1_t})$ for subgoal action $a_{1_t}$ involves passing subgoal $a_{1_t}$ to level 0 as the level 0's next goal. The level 0 goal-conditioned policy then has at most $n$ primitive actions

to achieve goal $a_{1_t}$. Thus for a 2-level agent, the top level goal-space $\mu_{\phi_1}$ and goal-conditioned policies $\pi_{\theta_1}$ will thus seek to learn the largest space of states reachable in $n$ subgoals, in which each subgoal can contain at most $n$ primitive actions. With this nested structure inspired by GCHRL, the time horizon of the computed empowerment can grow exponentially with the number of levels $k$, and no goal-conditioned policy is required to learn a long sequence of actions, making optimizing Goal-Conditioned Empowerment with RL more tractable. Figure 2 visualizes the proposed hierarchical architecture in one of our domains. Figure E in the Appendix provides some additional detail on the hierarchical architecture.

### 3.3 LIMITATIONS

The Hierarchical Empowerment framework has two significant limitations as currently constructed. The first is that the framework can only learn long horizon skills in domains in which the state space has large regions of reachable states that can be covered by $d$-dimensional goal space boxes. This limitation results from the assumption that the learned goal space takes the form of a uniform distribution across a $d$-dimensional subset of the state space. In the framework, the uniform goal space distribution will only expand to include achievable states. If regions adjacent to the goal space are not reachable, the goal space will not expand. Consequently, if the state space contains large regions of unreachable states (e.g., a pixel space or a maze), the framework will not be able to learn a meaningful space of skills.

The second limitation is that a model of the transition dynamics is required. The most important need for the model is to simulate the temporally extended subgoal actions so that (state, subgoal action, next state, reward) tuples can be obtained to train the higher level goal-conditioned policies with RL. However, the existing empowerment-based skill-learning approaches that learn the variational distribution $q_\psi(z|s_0, s_n)$ also require a model in large domains to obtain unbiased gradients of the maximum likelihood objective. The maximum likelihood objective requires sampling large batches of skills $z$ from large sets of skill starting states $s_0$ and then executing the *current* skill-conditioned policy to obtain $s_n$. In large domains, a model of the transition dynamics is needed for this to be feasible.

## 4 EXPERIMENTS

The purpose of our experiments is to evaluate the two claims of our framework. The first claim is that Goal-Conditioned Empowerment can more effectively compute empowerment (i.e., learn the largest space of distinct skills) than existing empowerment-based skill learning approaches that rely on a learned variational distribution. For this evaluation, we compare Goal-Conditioned Empowerment to DIAYN (Eysenbach et al., 2019), HIDIO (Zhang et al., 2021), and DADS (Sharma et al., 2020). The skill-learning objectives used by DIAYN and HIDIO are very similar to the typical empowerment-based skill-learning objective described in section 2.3. The key differences between DIAYN and HIDIO is the structure of the learned variational distributions. DIAYN learns the distribution $q_\psi(z|s_0, s_t)$ (i.e., each skill $z$ should target a specific state). In our implementation of HIDIO, we used the distribution that the authors note generally performed the best in their experiments: $q_\phi(z|s_0, a_{t-1}, s_t)$ (i.e., each skill targets an (action, next state) combination). DADS optimizes a variational lower bound to a related objective to Empowerment: $I(Z; S_1, S_2, \ldots, S_n|s_0)$ (i.e., each skill should target a trajectory of states). We discuss this objective in more detail in section H of the Appendix including how DADS also does not provide a strong signal for skills to differentiate when skills overlap. The second claim of our framework that our experiments seek to evaluate is that hierarchy is helpful for computing long-term empowerment. To assess this claim, we compare Hierarchical Empowerment agents that learn one, two, and three levels of skills.

### 4.1 DOMAINS

Although there are some emerging benchmarks for unsupervised skill learning (Laskin et al., 2021; Gu et al., 2021; Kim et al., 2021), we cannot evaluate Hierarchical Empowerment in these benchmarks due to the limitations of the framework. The benchmarks either (i) do not provide access to a model of the transition dynamics or (ii) do not contain state spaces with large contiguous reachable regions that can support the growth of $d$-dimensional goal space boxes. For instance, some

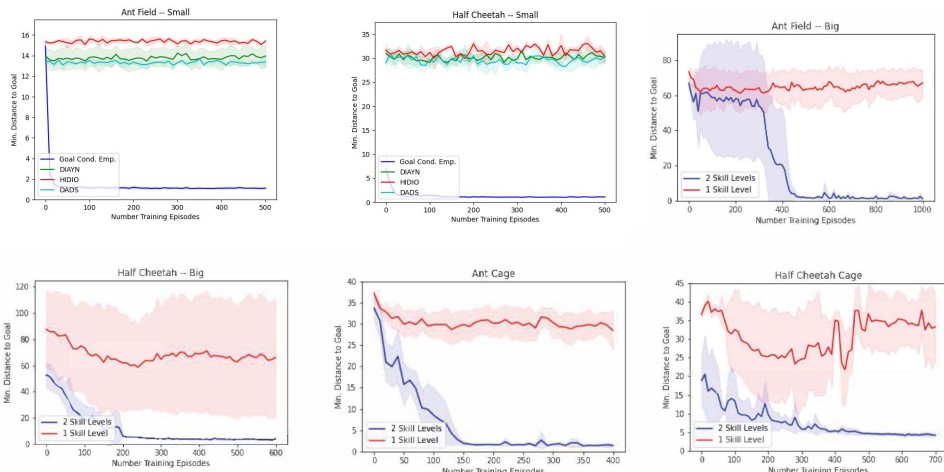

Figure 3: Figure provides phase 2 performance results. The minimum distance measurement is averaged over four random seeds. Each random seed averages 400 episodes. The error bars show one standard deviation.

frameworks involve tasks like flipping or running in which the agent needs to achieve particular combinations of positions and velocities. In these settings, there may be positions outside the current goal space the agent can achieve, but if the agent cannot achieve those positions jointly with the velocities in the current goal space, the goal space may not expand.

Instead, we implement our own domains in the popular Ant and Half Cheetah environments. We implement the typical open field Ant and Half Cheetah domains involving no barriers, but also environments with barriers. All our domains were implemented using Brax (Freeman et al., 2021b), which provides access to a model of the transition dynamics. In all domains, the learned goal space is limited to the position of the agent's torso ((x,y) positions in Ant and (x) in Half Cheetah). When limited to just the torso position, all the domains support large learned goal spaces. Also, please note the default half cheetah gait in Brax is different than the default gait in MuJoCo. We provide more detail in section J of the Appendix.

Each experiment involves two phases of training. In the first phase, there is no external reward, and the agent simply optimizes its mutual information objective to learn skills. In the second phase, the agent learns a policy over the skills to solve a downstream task. In our experiments, the downstream task is to achieve a goal state uniformly sampled from a hand-designed continuous space of goals. Additional details on how the first and second phases are implemented are provided in section G of the Appendix.

## 4.2 RESULTS

We present our results in a few ways. Figure 3 provides charts for the phase two performances for most of the experiments. Results of the three versus four level Ant experiment is provided in section A of the Appendix. We also provide a video presentation showing sample episodes of the trained agents at the following url: `https://www.youtube.com/watch?v=OOjtC30VjPQ`. In addition, we provide before and after images of the learned goal spaces in section C of the Appendix. Section K of the Appendix discusses some of the ablation experiments that were run to examine the limitations of the framework.

The experimental results support both contributions of the framework. Goal-Conditioned Empowerment substantially outperforms DIAYN, HIDIO, and DADS in the Ant Field – Small and Half Cheetah – Small domains. None of the baselines were able to solve either of the tasks. As we describe in section I of the Appendix, in our implementation of DIAYN we tracked the scale of the learned variational distribution $q_\phi(z|s_0, s_t)$. As we hypothesized, the standard deviation of the variational distribution did not decrease over time, indicating that the the skills were not specializing and

targeting specific states. On the other hand, per the performance charts and video, Goal-Conditioned Empowerment was able to learn skills to achieve goals in the phase two goal space.

The experimental results also support our claim that hierarchy is helpful for computing long-term empowerment. In both the two versus three level agents (i.e., one skill level vs. two skill level agents) and in the three versus four level comparison, the agents with the additional level outperformed, often significantly, the agent with fewer levels. In addition, the surface area of the skills learned by the four level Hierarchical Empowerment agents is significantly larger than has been reported in prior empowerment-based skill-learning papers. In our largest Ant domain solved by the four level agent, the phase 2 goal space with size 800x800 is over four orders of magnitude larger than the surface area reported in the DIAYN Ant task (4x4), and over 2 orders of magnitude larger than the area (30x30) partially achieved by Dynamics-Aware Unsupervised Discovery of Skills (DADS) (Sharma et al., 2020). Section B of the Appendix provides a visual of the differences in surface area coverage among Hierarchical Empowerment, DADS, and DIAYN. In terms of wall clock time, the longest goals in the 800x800 domain required over 13 minutes of movement by a reasonably efficient policy.

## 5 RELATED WORK

There are several categories of skill-learning closely related to our proposed framework (see section L of the Appendix for additional related work detail).

**Other Empowerment** There are some additional approaches that use empowerment or modified versions of empowerment to learn skills. These include SNN4HRL (Florensa et al., 2017), which augments a task reward with a mutual information reward to encourage more diverse skills. In addition, LSD (Park et al., 2022) optimizes a modified version of the skill channel mutual information to force the agent to learn more dynamic and long horizon skills.

**Automated Curriculum Learning** Similar to our framework are several methods that implement automated curricula of goal-conditioned RL problems. Maximum entropy-based curriculum learning methods (Pong et al., 2019; Pitis et al., 2020; Campos et al., 2020) separately optimize the two entropy terms that make up the mutual information of the skill channel. They generally try to learn a highly entropic generative model of the states visited and use the generative model as the goal space distribution (i.e., maximize $H(Z|s_0)$), while also separately training a goal-conditioned policy (i.e., minimize $H(Z|s_0, S_n)$). In addition, there are other automated curriculum learning methods that implement curricula of goal-conditioned RL problems based on learning progress (Florensa et al., 2018; Colas et al., 2019; Baranes & Oudeyer, 2013).

**Bonus-based Exploration.** Empowerment is related to another class of intrinsic motivation algorithms, Bonus-based Exploration. These approaches incentivize agents to explore by augmenting the task reward with a bonus based on how novel a state is. The bonuses these methods employ include count-based bonuses (Bellemare et al., 2016; Ostrovski et al., 2017; Lobel et al., 2023), which estimate the number of times a state has been visited and provide a reward inversely proportional to this number. Another popular type of bonus is model-prediction error (Schmidhuber, 1991; Pathak et al., 2017; Burda et al., 2019) which reward states in which there are errors in the forward transition or state encoding models. One key issue with bonus-based exploration methods is that it is unclear how to convert the agent's exploration into a large space of reusable skills.

## 6 CONCLUSION

We make two improvements to computing empowerment with Reinforcement Learning. The first contribution is a new variational lower bound on empowerment that combines the practical variational distribution from goal-conditioned RL with a learnable skill space — a combination that can make it easier to calculate short-term empowerment. The second improvement is a hierarchical architecture that makes computing long-term empowerment more tractable. We hope future work is able to overcome the framework's limitations and generalize the approach to larger classes of domains.

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

## A  THREE VS. FOUR LEVEL AGENT RESULTS

Figure 4 shows the phase 2 results for the 3 vs. 4 level agents (i.e., 2 vs. 3 skill levels) in the 800x800 Ant domain. 800x800 represents the dimensions of the phase 2 goal space, from which goals are uniformly sampled at the start of each phase 2 episode. As in the results showed earlier in Figure 3, the y-axis shows the average minimum distance to goal given a specific period of phase 2 training. The average is over (i) four phase 1 agents that were trained and (ii) 400 goals sampled from the phase 2 goal space for each phase 1 agent. A video of a trained four level agent in the 800x800 domain is available at the following url: `https://www.youtube.com/watch?v=Ghag1kvMgkw`.

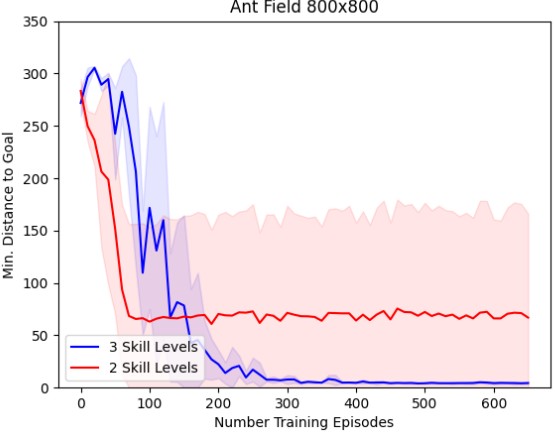

Figure 4: Phase 2 results of three vs. four level agents in 800x800 Ant domain.

## B  COMPARISON OF SURFACE AREA COVERAGE

Figure 5 compares the surface area coverage of the skills learned by Hierarchical Empowerment in the 800x800 domain to the surface area coverage in the published results of DIAYN (Eysenbach et al., 2019) and DADS (Sharma et al., 2020). The 800x800 domain is 40,000x and $\approx 700x$ larger than the surface area covered by the skills of DIAYN (4x4) and DADS (30x30), respectively. Note that the 30x30 surface area in the DADS results was only partially covered.

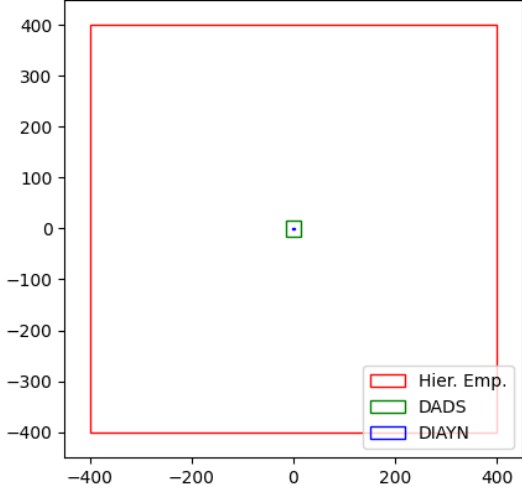

Figure 5: Comparison of (x,y) surface area coverage.

## C  GROWTH OF GOAL SPACES

Figures 6-10 provide visuals of the learned goal spaces before and after phase 1 for the agents that learn two and three levels of skills. The level 0, level 1, and level 2 (if applicable) goal spaces are shown by the blue, green, and red goal spaces, respectively. In all experiments, the goal spaces grow significantly from phase 1 of training. In the cage domains, one can also see how the learned goal spaces adjust to the barriers. In the ant cage domain, given that the button is in the northeast direction of the cage, the learned goal space expands more in the northeast direction. Also, in the half cheetah domain, the learned goal space is largely held within the cage.

**Ant Field – Big**

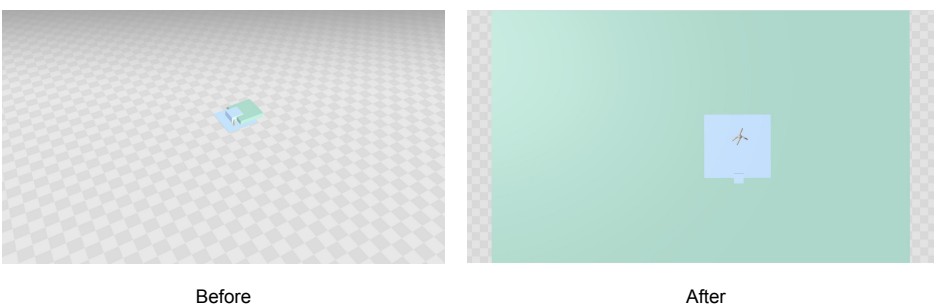

Before        After

Figure 6: Ant Field Big goal space growth.

## D  ALGORITHM AND IMPLEMENTATION DETAILS

In this section, we provide a detailed algorithm for how Hierarchical Empowerment is implemented. Before describing the algorithm, we note two small changes to the approach described so far that had a meaningful impact on our results. One change is that instead of having the goal space policy $\mu_\phi$ output the half widths of the uniform distribution, we have the policy output the $\log$ of the half widths. The reason for this change is that the entropy part of the reward $-\log p(z|s_0)$ only provides

**Half Cheetah – Big**

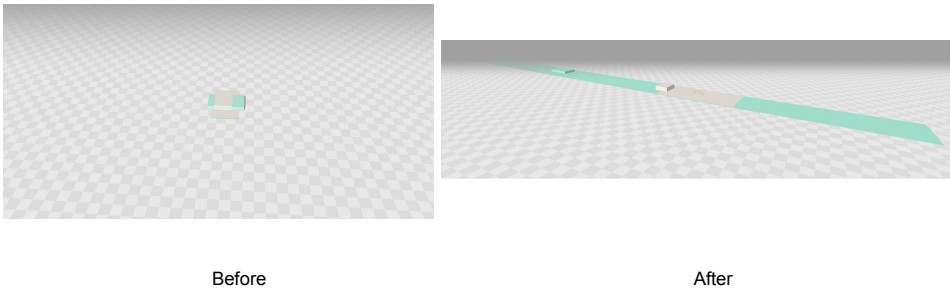

Before                                                                                      After

Figure 7: Half Cheetah Big goal space growth.

**Ant Cage**

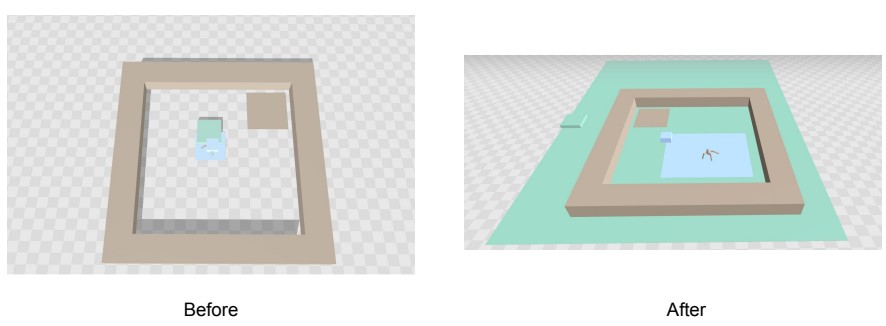

Before                                                                                      After

Figure 8: Ant Cage goal space growth.

a weak signal for the goal space to grow when the goal space reaches a certain size. This problem occurs because $-\log p(z|s_0)$ reduces to $\sum_{i=0}^{d} \log \mu^{w_i}(s_0) + \text{constant}$, in which $\mu^{w_i}(s_0)$ represents the half width of the $i$-th dimension of the goal space. Because $\log$ is a concave function, the entropy reward $\sum_{i=0}^{d} \log \mu^{w_i}(s_0)$ produces a diminishing incentive for the goal space to grow. In contrast, by having the goal space policy output the $\log$ of the half widths, the entropy portion of the reward becomes $\sum_{i=0}^{d} \mu^{w_i}(s_0) + \text{constant}$, which does not diminish as the goal space half widths $\mu^{w_i}(s_0)$ grow larger.

Another difference is that we optimize a regularized version of the goal-conditioned policy objective. Specifically, we optimize the objective

$$\mathbb{E}_{z\sim p_\phi(z|s_0), s\sim \rho_\theta(s|s_0,z)}[\log \mathcal{N}(h(s_0) + z; s, \sigma_0)], \tag{9}$$

in which $\rho_\theta(s|s_0,z)$ is the (improper) state visitation distribution: $\rho_\theta(s|s_0,z) = \sum_{t=0}^{n-1} p(s_0 \rightarrow s, t, z)$, in which is the $p(s_0 \rightarrow s, t, z)$ is the probability of moving from the initial state $s_0$ to state $s$ in $t$ actions when the goal-conditioned policy is pursuing goal $z$. The main difference between the regularized version of the objective is that the reward $r(s_t, z, a_t) = \mathbb{E}_{s_{t+1}\sim p(s_{t+1}|s_t,a_t)}[\log \mathcal{N}(h(s_0) + z|s_{t+1}, \sigma_0)]$ occurs at each of the $n$ steps instead of just the last step. We found this to produce more stable goal-conditioned policies.

Algorithm 1 provides pseudocode for Hierarchical Empowerment. The approach is implemented by repeatedly iterating through the $k$ levels of the agent and updating the goal-conditioned actor-critic and goal-space actor-critic.

Algorithm 2 provides the goal-conditioned actor-critic update function. The purpose of this function is to train the goal-conditioned policy to be better at achieving goals within and nearby the

**Half Cheetah Cage**

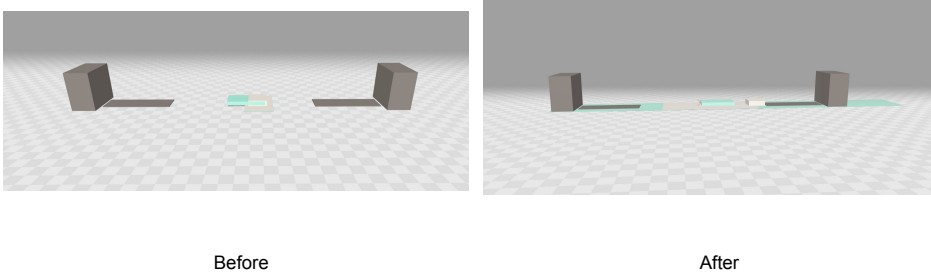

Before         After

Figure 9: Half cheetah cage goal space growth.

**Ant Field - 3 Skill Levels**

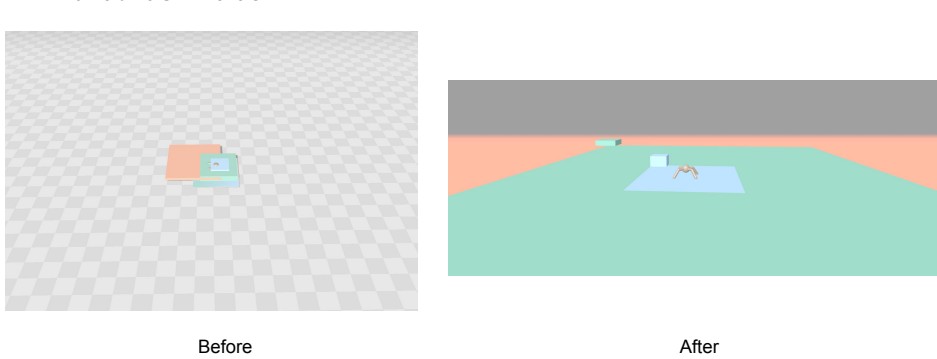

Before         After

Figure 10: Ant Field with 3 skill levels goal space growth.

current goal space. We assume a deterministic goal-conditioned policy, so the update function uses Deterministic Policy Gradient (DPG) (Silver et al., 2014) to update the actor.

Algorithm 3 provides the goal space actor-critic update function. The purpose of this algorithm is to enable the goal space policy $\mu_\phi$ to find the largest space of achievable goals. Given the deterministic goal space policy, this function also uses a DPG-style gradient.

Hierarchical Empowerment can be used with any skill start state distribution $p^{\text{level}}(s_0)$. We employ a process that requires no additional domain expertise. Beginning from a state sampled from the task's initial state distribution, our strategy repeats the following two step process for $N$ iterations. First, we sample a skill from top level goal space based on the agent's current state. Second, we greedily execute the goal-conditioned policy at each level until top level skill is complete. Prior to execution of the first action by the goal-conditioned policy at each level $i$, the current state is be saved into level $i$'s initial state buffer. The start state distribution for each level can then uniformly sample the start state buffer at that level. With this strategy, as the top level goal space grows, the start state distributions at all levels grows.

## E   HIERARCHICAL ARCHITECTURE DETAIL

Figure 11 provides some additional detail on the architectures of the goal-conditioned and goal-space policies. The figure on the left side shows the architecture for the goal-conditioned policy at level $k$, $\pi_{\theta_k}$. $\pi_{\theta_k}$ receives as input a state $s$ and goal $z$. The input is passed through the goal-conditioned policy neural network $f_{\theta_k}(s, z)$. $f_{\theta_k}(s, z)$ is then passed through a tanh activation function, which bounds $f_{\theta_k}(s, z)$ between $[-1, 1]$. The tanh output is then multiplied by bounds$_{k-1}$, which represents the halfwidths of the level $k - 1$ goal space. The goal space shift, shifts$_{k-1}$, is then added

---

**Algorithm 1** Hierarchical Empowerment

---

initialize $k$ goal-conditioned actor-critics $\{(\pi_{\theta_0}, Q_{\lambda_0}), \ldots, (\pi_{\theta_{k-1}}, Q_{\lambda_{k-1}}\}$
initialize $k$ goal space actor-critics $\{(\mu_{\phi_0}, R_{\omega_0}), \ldots, (\mu_{\phi_{k-1}}, R_{\omega_{k-1}})\}$
**repeat**
    **for** level $= 0$ **to** $(k-1)$ **do**
        $\theta_{\text{level}}, \lambda_{\text{level}} \leftarrow$ Update Goal-Conditioned Actor-Critic $(\theta_{\text{level}}, \lambda_{\text{level}}, \phi_{\text{level}})$
        $\phi_{\text{level}}, \omega_{\text{level}} \leftarrow$ Update Goal Space Actor-Critic$(\phi_{\text{level}}, \omega_{\text{level}}, \theta_{\text{level}})$
    **end for**
**until** convergence

---

**Algorithm 2** Update Goal-Conditioned Actor-Critic$(\theta_{\text{level}}, \lambda_{\text{level}}, \phi_{\text{level}})$

---

Initialize replay buffer $\mathcal{D}$
                                             $\triangleright$ Collect transition data
Sample skill state state $s_0 \sim p^{\text{level}}(s_0)$
Sample goal $z \sim s_0 + g(\epsilon \sim \mathcal{U}, \mu_{\phi_{\text{level}}}(s_0) + \text{noise})$    $\triangleright$ Reparameterization trick with noisy goal space
**for** step $t \in [0, \ldots, n-1]$ **do**
    Sample noisy action $a_t \sim \pi_{\theta_{\text{level}}}(s_t, z) + \text{noise}$
    Sample next state $s_{t+1} \sim p^{\text{level}}(s_{t+1}|s_t, a_t)$
    Compute reward $r_{t+1}(s_{t+1}, z) = \log \mathcal{N}(h(s_0) + z; s_{t+1}, \sigma_0)$
    Determine discount rate: $\gamma_{t+1} = 0$ if $d(s_{t+1}, z) < \varepsilon$ else $\gamma_{t+1} = \gamma$
    Add transition to replay buffer: $\mathcal{D} \cup \{(s_t, z, a_t, r_{t+1}, s_{t+1}, \gamma_{t+1})\}$
**end for**
**for** gradient step $s \in [0, \ldots, S-1]$ **do**
    Sample batch of transitions $B \subseteq \mathcal{D}$    $\triangleright B = \{(s_t^i, z^i, a_t^i, r_{t+1}^i, s_{t+1}^i, \gamma_{t+1}^i)\}, 0 \leq i \leq |B|$
                                                      $\triangleright$ Update critic
    Compute target Q-Values: $\text{Target}^i = r_{t+1}^i + \gamma_{t+1}^i Q_{\lambda_{\text{level}}}(s_{t+1}^i, z^i, \pi_{\theta_{\text{level}}}(s_{t+1}^i, z^i))$
    Compute critic loss: $L_{\text{critic}}(\lambda_{\text{level}}, B) = \frac{1}{|B|} \sum_i (Q_{\lambda_{\text{level}}}(s_t^i, z^i, a_t^i) - \text{Target}^i)^2$
    Update critic parameters: $\lambda_{\text{level}} \leftarrow \lambda_{\text{level}} - \alpha \nabla_{\lambda_{\text{level}}} L_{\text{critic}}(\lambda_{\text{level}}, B)$
                                                      $\triangleright$ Update actor
    Compute actor loss: $L_{\text{actor}}(\theta_{\text{level}}, \lambda_{\text{level}}, B) = \frac{1}{|B|} \sum_i -Q_{\lambda_{\text{level}}}(s_t^i, z^i, \pi_{\theta_{\text{level}}}(s_t^i, z^i))$
    Update actor parameters: $\theta_{\text{level}} \leftarrow \theta_{\text{level}} - \alpha \nabla_{\theta_{\text{level}}} L_{\text{actor}}(\theta_{\text{level}}, \lambda_{\text{level}}, B)$
**end for**
**return** $\theta_{\text{level}}, \lambda_{\text{level}}$

---

**Algorithm 3** Update Goal Space Actor-Critic$(\phi_{\text{level}}, \omega_{\text{level}}, \theta_{\text{level}})$

---

Initialize replay buffer $\mathcal{D}$
                                             $\triangleright$ Collect transition data
Sample skill start state $s_0 \sim p^{\text{level}}(s_0)$
Sample epsilon $\epsilon \sim \mathcal{U}$
Sample noisy goal space action $a \sim \mu_{\phi_{\text{level}}}(s_0) + \text{noise}$
Compute goal with reparameterization trick: $z = g(\epsilon, a)$        $\triangleright z$ is desired change from $s_0$
Sample skill-terminating state $s_n \sim p_{\theta_{\text{level}}}(s_n|s_0, z)$
Compute reward $r = \log \mathcal{N}(h(s_0) + z; s_n, \sigma_0) - \log p(z|a)$
Add transition to replay buffer: $\mathcal{D} \cup \{(s_0, \epsilon, a, r)\}$
**for** gradient step $s \in [0, \ldots, S-1]$ **do**
    Sample batch of transitions $B \subseteq \mathcal{D}$                $\triangleright B = \{(s_0^i, \epsilon^i, a^i, r^i)\}, 0 \leq i \leq |B|$
                                                      $\triangleright$ Update critic
    Compute critic loss: $L_{\text{critic}}(\omega_{\text{level}}, B) = \frac{1}{|B|} \sum_i (R_{\omega_{\text{level}}}(s_0^i, \epsilon^i, a^i) - r^i)^2$
    Update critic parameters: $\omega_{\text{level}} \leftarrow \omega_{\text{level}} - \alpha \nabla_{\omega_{\text{level}}} L_{\text{critic}}(\omega_{\text{level}}, B)$
                                                      $\triangleright$ Update actor
    Compute actor loss: $L_{\text{actor}}(\phi_{\text{level}}, \omega_{\text{level}}, B) = \frac{1}{|B|} \sum_i -R_{\omega_{\text{level}}}(s_0^i, \epsilon^i, \mu_{\phi_{\text{level}}}(s_0))$
    Update actor parameters: $\phi_{\text{level}} \leftarrow \phi_{\text{level}} - \alpha \nabla_{\phi_{\text{level}}} L_{\text{actor}}(\phi_{\text{level}}, \omega_{\text{level}}, B)$
**end for**
**return** $\phi_{\text{level}}, \omega_{\text{level}}$

---

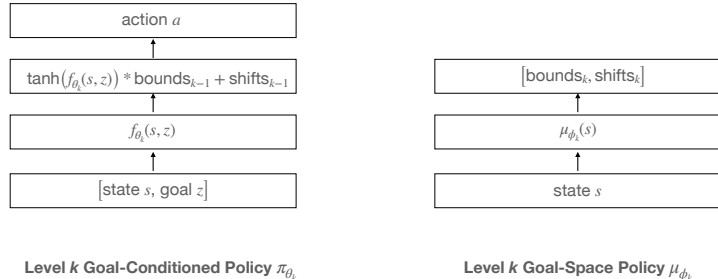

Figure 11: Policy architectures for level $k$ of the hierarchical agent.

to the resulting output. $[\text{bounds}_{k-1}, \text{shifts}_{k-1}]$ can be sampled from the goal-space policy $\mu_{\phi_{k-1}}$ at level $k-1$. Note that for the bottom level policy that outputs primitive actions in the range $[-1, 1]$, $\text{bounds}_{-1} = 1$, and $\text{shifts}_{-1} = 0$.

The right side of the figure shows the architecture for a level $k$ goal-space policy. The goal-space policy $\mu_{\phi_k}$ at level $k$ takes as input a state $s$ and outputs the bounds (i.e., the halfwidths) and shifts of each dimension of the goal space. Note that in phase 2 of our experiments when an additional level is added to the agent in order to learn a policy over the skills, this added level only includes a goal-conditioned policy and not a goal-space policy.

## F  GOAL SPACE CRITIC OBJECTIVE

The objective for training the goal-space critic is

$$J(\omega) = \mathbb{E}_{(s_0, \epsilon, a) \sim \beta}[(R_\omega(s_0, \epsilon, a) - \text{Target})^2], \tag{10}$$
$$\text{Target} = \mathbb{E}_{s_n \sim p(s_n|s_0, z)}[\log \mathcal{N}(h(s_0) + z; s_n, \sigma_0) - \log p(z|s_0)].$$

In Equation 10, the learned critic is trained to be closer to a target value using $(s_0, \epsilon, a)$ tuples from a buffer $\beta$, in which $s_0$ is a skill start state, $\epsilon$ is a unit uniform random variable from the reparameterization trick, $a$ is a goal space action (i.e., the vector of location-scale parameters for the uniform distribution goal space). The target is obtained by first executing the skill $z = g(\epsilon, a)$ and then sampling the terminal state $s_n$. The target is then the sum of (i) the $\log \mathcal{N}(h(s_0) + z; s_n, \sigma_0)$, which measures how effective the goal-conditioned policy is at achieving the goal state $h(s_0) + z$ and (ii) $-\log p(z|s_0)$, which comes from the entropy reward that was folded in.

## G  KEY HYPERPARAMETERS

Tables 1 and 2 show the key hyperparameters for phase 1 and phase 2 training, respectively. In Table 1, $k$ refers to the number of skills levels in each agent (note than this does not include the phase 2 policy). $n$ represents the maximum number of actions the goal-conditioned policy at each level has to reach its goal. For instance, for a $k = 2$ agent with $n = [20, 10]$, this means the level 0 (i.e., the bottom level) goal-conditioned policy has at most 20 actions to reach its goal, while level 1 has at most 10 actions. $\sigma_0^{\text{gc}}$ and $\sigma_0^{\text{gs}}$ are the standard deviations of the fixed variance Gaussian variational distributions used for the reward functions for the goal-conditioned and goal space policies, respectively. The larger standard deviation for training the goal space policy provides some additional flexibility, helping goal space growth. $\varepsilon$ is the goal threshold. The goal-conditioned policy terminates when the agent is within $\varepsilon$ of the goal.

The last column in 1 lists the number of epochs of training in phase 1. For the goal-conditioned actor-critics, each epoch of training in phase 1 consists of 10 iterations of Algorithm 2, in which the number of gradient steps per iteration $S = 50$. For the goal space actor-critics, each epoch of training consists of a single iteration of Algorithm 3, in which the number of gradient steps per iteration $S = 10$.

Table 2 lists key hyperparameters for phase 2 of training. $n$ provides the maximum number of attempts the phase 2 goal-conditioned policy has to achieve its goal. $\varepsilon$ provides the phase 2 goal

Table 1: Phase 1 Hyperparameter Selections

| Domain | $k$ | $n$ | $\sigma_0^{\text{gc}}$ | $\sigma_0^{\text{gs}}$ | $\varepsilon$ | Phase 1 Epochs |
|---|---|---|---|---|---|---|
| Ant Field Small | 1 | 20 | 0.4 | 1.75 | 0.6 | 1220 |
| Ant Field Big | 1 | 80 | 0.4 | 1.75 | 0.6 | 1900 |
| Ant Field Big | 2 | $[20, 10]$ | $[0.4, 0.8]$ | $[1.75, 3.5]$ | $[0.6, 1.2]$ | 1900 |
| Ant Field XL | 2 | $[50, 40]$ | $[0.4, 0.8]$ | $[1.75, 3.5]$ | $[0.6, 1.2]$ | 1450 |
| Ant Field XL | 3 | $[20, 10, 10]$ | $[0.4, 0.8, 1.6]$ | $[1.75, 3.5, 7.0]$ | $[0.6, 1.2, 2.4]$ | 1450 |
| Half Cheetah Small | 1 | 20 | 0.3 | 2.0 | 1.0 | 1000 |
| Half Cheetah Big | 1 | 200 | 0.3 | 2.0 | 1.0 | 1300 |
| Half Cheetah Big | 2 | $[20, 10]$ | $[0.3, 3.0]$ | $[2.0, 10.0]$ | $[1.0, 2.0]$ | 1300 |
| Ant Cage | 1 | 80 | 0.4 | 1.75 | 0.6 | 1900 |
| Ant Cage | 2 | $[20, 10]$ | $[0.4, 0.8]$ | $[1.75, 3.5]$ | $[0.6, 1.2]$ | 1900 |
| Half Cheetah Cage | 1 | 200 | 0.3 | 2.0 | 1.0 | 980 |
| Half Cheetah Cage | 2 | $[20, 10]$ | $[0.3, 3.0]$ | $[2.0, 10.0]$ | $[1.0, 2.0]$ | 980 |

Table 2: Phase 2 Hyperparameter Selections

| Domain | $k$ | $n$ | $\varepsilon$ | Goal Space Length (each dim) |
|---|---|---|---|---|
| Ant Field Small | 1 | 10 | 1.8 | 40 |
| Ant Field Big | 1 | 25 | 2.4 | 200 |
| Ant Field Big | 2 | 10 | 2.4 | 200 |
| Ant Field XL | 2 | 10 | 9.6 | 800 |
| Ant Field XL | 3 | 10 | 9.6 | 800 |
| Half Cheetah Small | 1 | 10 | 3.0 | 95 |
| Half Cheetah Big | 1 | 10 | 10.0 | 450 |
| Half Cheetah Big | 2 | 10 | 10.0 | 450 |
| Ant Cage | 1 | 25 | 2.4 | 100 |
| Ant Cage | 2 | 10 | 2.4 | 100 |
| Half Cheetah Cage | 1 | 10 | 10.0 | 150 |
| Half Cheetah Cage | 2 | 10 | 10.0 | 150 |

threshold. The last column provides the length of each dimension of the goal space. In the Ant domains, the goal space is two-dimensional, while in the Half Cheetah domains, the goal space is one-dimensional. Also, given that it is easier to move forward than backward in Half Cheetah, we do slightly shift the goal space forward in the half cheetah domains. In all domains, at the start of each episode in phase 2, a goal is uniformly sampled from the phase 2 goal space.

# H  DADS OBJECTIVE

DADS optimizes a variational lower bound to a related objective to Empowerment: $I(Z; S_1, S_2, \ldots, S_n | s_0)$ (i.e., each skill should target a trajectory of states). The authors observe that a variational lower bound on this objective is equivalent to a reinforcement learning problem in which the reward is $r(s_t, a_t, s_{t+1}) = \log \frac{q_\phi(s_{t+1}|s_0,s_t,z)}{p(s_{t+1}|s_0,s_t)}$, in which $q_\phi(s_{t+1}|s_0, s_t, z)$ is a learned variational distribution that seeks to learn the distribution of next states and the marginal $p(s_{t+1}|s_0, s_t)$ is estimated with the average $(\sum_{i=1}^{L} q_\phi(s_{t+1}|s_0, s_t, z_i))/L$ for $z_i \sim p(z)$. This reward will be higher if skills $z$ target unique regions of the state trajectory space as the variational distribution $q_\phi(s_{t+1}|s_0, s_t, z)$ will be high and the marginal probability $p(s_{t+1}|s_0, s_t)$ will be low. The problem again with this strategy is that in times when the skills overlap $p(s_{t+1}|s_0, s_t)$ will generally be high resulting in rewards that are generally low, which in turn produces little signal for the skills to differentiate.

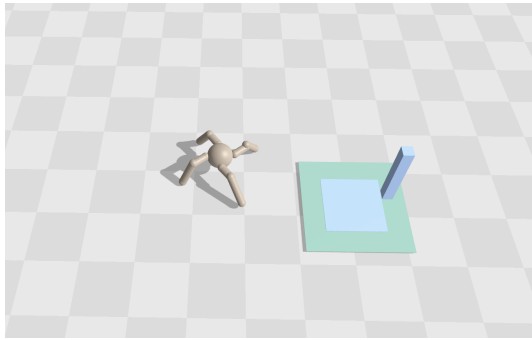

Figure 12: Visualization of DIAYN's overlapping skills.

## I  DIAYN VISUALIZATION

After phase 1 of training in our DIAYN implementation there was still significant overlap among the skills in the skill space. We can infer this outcome by visualizing the learned variational distribution. Figure 12 provides a visualization of what the learned variational distribution typically looked like after phase 1 of training. The green rectangle shows the fixed skills space in our implementation DIAYN, which is the continuous space $[-1, 1]^2$. The blue rectangular prism, is the current skill that is being executed in this image. The blue rectangle shows one standard deviation of the learned variational distribution which is a diagonal Gaussian distribution. This blue rectangle thus shows the probability distribution over skills given the tuple (skill start state, current state). Per the size of the single standard deviation, the learned variational distribution covers most of the skill space, meaning a large space of skills could have led to the current state (i.e., the skills are not differentiated).

## J  BRAX HALF CHEETAH GAIT

At the time of writing, the default forward gait from the Brax Freeman et al. (2021a) version of Half Cheetah is noticeably different than the typical MuJoCo Todorov et al. (2012) gait. Instead of the half cheetah using its feet to run, the half cheetah will propel itself forward by pushing off it back knee. We provide a video of the default forward gait at the following link: `https://www. youtube.com/watch?v=4IKUf19GONw`. Our implementation of half cheetah exacerbates this issue as we increase the number of physics steps per action so that the agent can move a reasonable amount in a smaller number $N$ actions.

## K  ABLATION EXPERIMENTS

In addition to the domains described in section 4.2 that were designed to support the goal space growth of Hierarchical Empowerment agents, we also implemented some ablation experiments that confirmed the current limitations of the framework. These includes various types of mazes. For instance, in one experiment we implemented a maze in the shape of an "H" with a horizontal hallway in the middle and two vertical hallways at each end of the horizontal hallway. The episodes start with the agent in the center of the horizontal hallway. In this experiment and in the other mazes we implemented, the agent was not able to learn skills that extend beyond the initial horizontal hallway. The likely reason is that in order to further extend the goal space to the vertical hallways, the goal space would need to include unreachable states that are north and south of the initial horizontal hallway. Although the entropy component of the goal-space reward $(-\log p(z|s_0))$ is larger when parts of the vertical hallways are included because the goal space is larger, the other component of the reward that measures how well goals within the goal space are achieved $(\log \mathcal{N}(h(s_0) + z; s_n, \sigma_0))$ is lower because $s_n$ is further from the goal $h(s_0) + z$.

## L    RELATED WORK DETAIL

There is an extensive history of skill-learning approaches (Parr & Russell, 1997; Sutton et al., 1999b; Dietterich, 1999; Stolle & Precup, 2002; Simsek et al., 2005; Konidaris & Barto, 2009; Bacon et al., 2017). Here we provide some additional details not mentioned in the body of the paper due to space constraints.

**Automated Curriculum Learning** Similar to our framework are several methods that implement automated curricula of goal-conditioned RL problems. Maximum entropy-based curriculum learning methods (Pong et al., 2019; Pitis et al., 2020; Campos et al., 2020) separately optimize the two entropy terms that make up the mutual information of the skill channel. That is, they try to learn a goal space with large entropy $H(Z|s_0)$ while also learning a goal-conditioned policy to reduce the conditional entropy $H(Z|s_0, S_n)$. In Skew-Fit (Pong et al., 2019), the distribution of goals tested at each phase of the curriculum is determined by a generative model, which is trained to model the distribution of states visited by the agent's current set of skills. The generative model is skewed to be closer to a uniform distribution so that states that are new to the agent are tested more frequently. In MEGA (Pitis et al., 2020), the set of goal states tested is determined using a learned density model of the states that have been visited previously. MEGA selects states with low probabilities per the density function so the states along the frontier of the visited state space can be tested more often. In both Skew-Fit and MEGA, a goal-conditioned policy is simultaneously but separately trained to minimize $H(Z|s_0, s_n)$. Similarly, EDL (Campos et al., 2020) uses a three stage process to separately optimize the entropies $H(S_n|s_0)$ and $H(S_n|s_0, Z)$. In the exploration stage, an exploration algorithm (Lee et al., 2019) is used to try to learn a uniform distribution over the state space. In the discover phase, the state space is encoded into a latent skill space in order to try to produce a high entropy skill space. In the learning phase, a goal-conditioned policy is trained to reach goal latent states.

There are two concerns with this approach of separately optimizing the goal space and goal-conditioned policy. One issue is that it is unlikely to scale to more realistic settings involving some randomness where many visited states are not consistently achievable. When applying the separate optimization approach to this more realistic setting setting, the skill space will still grow, producing higher entropy $H(Z|s_0, s_n)$. However, because many of the goal states are not achievable, the conditional entropy $H(Z|s_0, s_n)$ will also grow, resulting in a lower empowerment skill space. A more promising approach is to jointly train the two entropy terms, as in our approach, so that the goal space only expands to include achievable goals. However, changes will need to be made to our framework to handle settings in which states are not achievable. A second issue with these methods is that is unclear how hierarchy can be integrated to better handle long horizon tasks.

Another class of automated curriculum learning methods implements curricula of goal-conditioned RL problems based on learning progress. Goal GAN (Florensa et al., 2018) trains a GAN (Goodfellow et al., 2014) to output goals in which the agent has made an intermediate level of progress towards achieving. CURIOUS (Colas et al., 2019) and SAGG-RIAC (Baranes & Oudeyer, 2013) output goals in which the agent has shown improvement.

**Bonus-based Exploration.** Empowerment is related to another class of intrinsic motivation algorithms, Bonus-based Exploration. These approaches incentivize agents to explore by augmenting the task reward with a bonus based on how novel a state is. The bonuses these methods employ include count-based bonuses (Bellemare et al., 2016; Ostrovski et al., 2017; Lobel et al., 2023), which estimate the number of times a state has been visited and provide a reward inversely proportional to this number. Another popular type of bonus is model-prediction error (Schmidhuber, 1991; Pathak et al., 2017; Burda et al., 2019) which reward states in which there are errors in the forward transition or state encoding models. One issue with bonus-based exploration methods is that it is unclear how to convert the agent's exploration into a large space of reusable skills. A second issue, particularly with the model-prediction error bonuses, is that these methods can be difficult to implement as it is unclear how frequently to update the model that determines the bonus relative to the exploration policy.

