# OpenReview forum: "Hierarchical Empowerment: Towards Tractable Empowerment-Based Skill Learning"
_ICLR.cc/2024/Conference — Submitted to ICLR 2024_

### Official Review · Reviewer_6xWS · 2023-10-23

**Soundness:** 2 fair
**Presentation:** 2 fair
**Contribution:** 2 fair
**Rating:** 3
**Confidence:** 4

**Summary:**

This paper tackles hierarchical goal-conditioned RL through the lens of empowerment. The main contributions are twofold: (1) a novel objective to jointly train a goal proposal distribution and a goal-reaching policy based on empowerment with a reparameterization trick and (2) a hierarchical architecture that combines multiple goal-reaching policies in a hierarchical manner. The authors evaluate their method in several Ant/HalfCheetah goal-reaching tasks, showing that their hierarchical architecture enables long-horizon goal reaching.

**Strengths:**

- The reparameterized variational (goal) empowerment objective in Eq. (7) seems novel and intriguing.
- The proposed hierarchical structure does expand the area of reachable goals.

**Weaknesses:**

## Weaknesses
- One major weakness of the proposed approach is that it assumes that the goal distribution $p_\phi(z|s_0)$ is a uniform distribution over a *box*. This significantly limits the scalability of the method, because the set of feasible goals can form an arbitrary shape even in simple environments (e.g., mazes).
- Another major weakness of this work is its limited empirical evaluation. For me, the proposed method is much closer to previous unsupervised GCRL methods (e.g., Skew-Fit, EDL, LEXA, etc.) than empowerment-based methods (e.g., DIAYN, DADS, HIDIO, etc.) given that (1) the latter uses a separate latent skill space $Z$ that is not "grounded" to the state space and (2) the former has the two-stage structure of goal proposal and goal reaching, similarly to the proposed approach. However, the authors only compare their method with empowerment-based approaches, which have already been known to struggle to reach distant goals (as shown in multiple prior works, such as EDL and LSD). I believe comparisons with unsupervised GCRL methods are necessary to assess the effectiveness of this method.
- Moreover, the experiments are only conducted in Ant and HalfCheetah environments with $x$-$y$ goal spaces without obstacles. It is unclear how the proposed method performs in more complex environments (e.g., AntMaze) or other types of environments (e.g., Fetch manipulation environments), both of which are standard benchmarks for goal-conditioned RL.
- The proposed method seems to assume access to the ground-truth transition dynamics function, which prevents its applicability to general environments.
- In terms of writing, I feel Section 3.1 is not sufficiently clear. Please see my questions below.

## Minor issues
- "The major problem with using empowerment for skill learning is that mutual information is difficult to optimize.": Do the authors have any supporting evidence for this claim?
- In Equations (4)-(5), I would suggest using $I^{GCE}(Z; S_n|s_0)$ to denote the mutual information (not its variational bound) and replacing "$=$" in Equation (5) with "$\geq$", following the convention.
- "However, the existing empowerment-based skill-learning approaches that learn the variational distribution $q_\psi(z | s_0, s_n)$ also require a model in large domains to obtain unbiased gradients of the maximum likelihood objective.": I'm not sure if this can justify the second limitation because (1) albeit biased, previous empowerment-based approaches can still work without a model and (2) we can relabel $z$ as in P-HER (Choi et al., 2021) to make it (much) less biased with respect to the current skill discriminator. In this regard, I think "First, similar to other empowerment-based skill-learning methods, our framework assumes the agent has access to a model of the environment’s transition dynamics." is a bit misleading.
- I would suggest using a higher-resolution image (or a vectorized pdf file) for Figure 3.


In summary, while I do think the reparameterized empowerment objective is intriguing, the limitations of the proposed approach seem to outweigh its strengths, and the empirical results are too weak to support the claimed benefits of the method.

**Questions:**

- I found Sec 3.1 to be a bit confusing. Specific questions:
    - What is $g$? Is it just an affine function?
    - Is $h$ a learnable function, or just a projection to the goal space?
    - "instead of sampling the skill z from the fixed variational distribution": Why do we need to sample $z$ from the variational distribution? Isn't $z$ deterministically determined by $\epsilon$ and $\mu_\phi(s_0)$?
- Where exactly does the proposed method use the ground-truth dynamics model? Is it required only for training higher-level policies, or is it used by the low-level policy as well?
- The authors argue that the proposed method can reach much more distant goals (800x800) than DADS (30x30). However, I'm not sure if this is an apples-to-apples comparison given that (1) their environments are different (MuJoCo vs Brax), and (2) their episode lengths may be (very) different. What is the maximum episode length used in this paper?

---

> ### Author Response · Authors · 2023-11-20
>
> Thank you for your feedback.
>
> ### Missing comparisons to Unsupervised Goal-Conditioned Algorithms (MEGA, Skew-fit, EDL, etc.)
>
> We disagree that a comparison with the unsupervised GCRL algorithms (MEGA, Skew-fit, EDL) is necessary.  Given the importance of Empowerment as a way to find the space of reachable states and thus quantify the size of an agent’s skillset, there needs to be algorithms that optimize the full Empowerment objective.  But MEGA, Skew-fit, and EDL do not optimize the full Empowerment objective.  Rather they optimize two different objectives.  In one process, they optimize $H(Z|s_0)$ or $H(S_N|s_0)$ (e.g., training an autoencoder on visited states to obtain a goal space $p(z|s_0)$).  At the same time, but in a second separate process, they train and goal-conditioned policy using goals sampled from the goal space to minimize $H(Z|s_0,S_N)$ or $H(S_N|s_0,Z)$.  The clear failure case for this approach is when there is randomness and the goal states from the goal space are not consistently achievable.  In this situation, this strategy may not calculate empowerment accurately (i.e., find the largest space of reachable states.) because the skill space is not optimized to learn skills that are consistently achievable. Thus in our baselines, we just considered algorithms that optimize the full Empowerment objective.
>
> ### “Albeit biased, previous empowerment-based approaches can work without a model.”
>
> It’s difficult to compare our empowerment-based work with earlier results because both our (a) skill space learned for each skill start state and (b) the distribution of skill start states seem to be vastly larger than prior work (i.e., our $p(z|s_0)$ and $p(s_0)$ are significantly more entropic), particularly in our larger settings.  Storing and sampling biased $(s_0,z,s_N)$ tuples from a replay buffer seems unlikely to scale to very large domains.  For instance, given that the replay buffer contains stale (i.e, biased) data, you can only use the recent past.  But then in large domains, much of the state space may not be continually visited so if you cut off your replay buffer, then your empowerment optimization may be missing those reachable states.
>
>
> ### Section 3.1 Clarifications
>
> $g$ represents the reparameterization trick.  For our uniform goal space distributions, each dimension of $z_i = -\text{halfwidth (output by goal space policy)} + 2*\text{halfwidth}*\text{eps}$, in which eps is a sample from a unit uniform random variable.  $h$ is just a projection to the goal space (e.g., the (x,y) coordinates from the state).
>
> The line "instead of sampling the skill z from the fixed variational distribution" was just referring to how the log of the variational posterior $\log q(z|s_0,s_N)$ is calculated, not how the skill $z$ is sampled.  Instead of computing $\log N(z; \text{mean}=s_N,\text{std}=\sigma_0)$ we use  $\log N(z + h(s_0);\text{mean}=s_N,\text{std}=\sigma_0)$.  This causes the skill $z$ to reflect desired change in state $s_0$ (e.g., desired change in the x and y coordinates from the skill start state $s_0$) and not actual actual states.  This is helpful because it initially centers the learned goal space on the start state $s_0$.
>
>
> ### Where is the model used and where is it required?
>
> We use the model to help generate transitions for all the goal-conditioned and goal-space policies.  The goal-conditioned policies need unbiased (prior state,action,next state) tuples and the goal-space policies need unbiased (skill start state, goal space action, epsilon, skill end state) tuples.  The model is not required for the base level goal-conditioned policy because a replay buffer of (prior state, action, next state) transitions would be unbiased.  A model is also technically not required for the goal space policies because the Q function from the goal-conditioned policies at the skill start states could be used as the critic for the goal space policy but we chose to use the model to generate transitions to train a separate goal space critic.
>
> ### State coverage is not apples-to-apples with DIAYN and DADS
>
> For our largest setting, our episode lengths were 2000 actions, which is likely significantly larger than in DIAYN and DADS papers.  However, our implementation of DIAYN and DADS could not learn skills to achieve goals in a 20x20 box so it seems very unlikely the extra time would make a noticeable difference in their state coverage.  In regards to the simulator difference, the authors of the Ant domain in Brax note that it was derived from the MuJoCo Ant domain.  We also have experience implementing Ant in MuJoCo and the domains seem similar so it seems unlikely our results would be too different in MuJoCo.

---

> > ### Comment · Reviewer_6xWS · 2023-11-21
> >
> > Thank you for the responses. I have carefully read them, but most of my initial concerns remain unresolved and I would maintain my original rating this time.
> >
> > I still think that the lack of comparison with unsupervised goal-conditioned RL methods is one of the major weaknesses of this method (in addition to the first major concern regarding limited applicability mentioned in my original review). The proposed method is, in fact, quite different from previous empowerment-based methods, such as DIAYN, DADS, and HIDIO, because the latter do not "ground" $z$ to real states (but treat it as a latent variable that can have arbitrary values), while this method essentially treats $z$ as goals. This "grounding" makes this method much more similar to previous goal-conditioned methods. I agree that previous goal-conditioned methods (Skew-Fit, EDL, etc.) may not optimize the full empowerment objective, but neither does this method, given that this method uses a *very* specific type of variational distribution (which treats $z$ as goals) and thus may not fully approximate the empowerment objective. More importantly, these unsupervised GCRL methods share the exactly same problem setting and have conceptually very relevant objectives. As such, I believe a comparison with these methods is necessary to assess the effectiveness of the proposed method.

---

### Official Review · Reviewer_2VaP · 2023-10-31

**Soundness:** 2 fair
**Presentation:** 2 fair
**Contribution:** 2 fair
**Rating:** 5
**Confidence:** 4

**Summary:**

The paper introduces a new framework, Hierarchical Empowerment, as a new method for calculating empowerment, or the mutual information between skills and states. The framework propose to use an objective from goal-conditioned RL as a variational lower bound on the empowerment objective. The paper also propose to define hierarchies at multiple levels (instead of the usual two-level hierarchy) as a way to address exploration and credit assignment. The authors evaluate their method on the Brax simulator using Ant and Half Cheetah, with and without barriers. Some qualitative results are also presented.

**Strengths:**

* The authors go beyond the standard two-level hierarchy that is typically used in HRL
* The paper tries to provide a new solution for maximizing empowerment, which is known to be difficult to optimize
* The paper introduces more compositional versions of the base Ant and Half Cheetah environments. Such compositionality is key for HRL to be succesful.

Maximizing the mutual information between states and skills has been the core of a long series of paper. In spirit the objective is intuitive, but it rarely leads to diverse skill unless stronger assumptions are made. This has been recently shown very well in [1].

Most of recent work in HRL only leverages a two-level hierarchy, from this perspective the paper investigate a setting that is rarely investigated.

**Weaknesses:**

* The baselines presented are outdated. Moreover the quantitative results show results for only 4 seeds
* The presentation of the paper could be greatly improved. Some sections feel unnecessary and some claims do not seem correct
* The qualitative evaluation does not show any in-depth analysis

The paper compares to baselines that are from a few iterations of research ago. There have been recent advances in maximizing empowerment. Moreover, the general evaluation is too narrow: it only focuses on empowerment-based HRL methods. [2] has recently establishes a new state-of-the-art across many domains, where it greatly improves upon empowerment-based methods. Without such baselines it is impossible to evaluate the merit of the method.

The presentation of the paper could be improved: for example, what is the role of section 2.2? Is it absolutely necessary for understanding the contributions? If not it should go in the appendix. More central to the method itself, the paper claims that a new objective is derived for empowerment. However equation (3) is essentially the same objective that we have seen across the literature on empowerment-based methods. It seems like the only difference is the way the variational distribution q is parametrized. In itself this is not a bad thing, but coupled with strong claims that "it explicitly encourages skills to target specific states" and that the objective itself is "new" does not help the paper's quality.

Moreover, despite claims that the method learns larger spaces and diverse skills, the qualitative evaluation is very limited, for example Figure 5 of Section B. Much more work is needed in order to convince the reader that this is indeed the case.

**Questions:**

One of the contributions of the method is to learn multiple levels of hierarchy (this was investigated [3] but the paper is not cited), however there are no experiments that separate the effect of the objective for empowerment with the multiple levels. Which one provides the gains?

"The benchmarks either (i) do not provide access to a model of the transition dynamics" Why is the model of the transition dynamics needed?

"The entropy term H ϕ(Z | s 0) encourages the goal space policy µ ϕto output larger goal spaces." This surely can't be right, as the goal space is predefined. Perhaps the authors mean that the distribution of generated goals will cover more space?

"q(· | s_0, s_n) is a diagonal Gaussian distribution with mean s_n and a fixed standard deviation s_0 set by the designer." What happens when the states are pixels?

"skills are an instance of noisy channels from Information Theory" This seems a bit strong, perhaps it can be interpreted but it doesn't not have to be an instance of something else.

"hand-crafted goal space may only achieve a loose lower bound on empowerment" Little justification for this is given.



====================================================================================

[1] Controllability-Aware Unsupervised Skill Discovery. Park et al. 2023

[2] Deep Laplacian-based Options for Temporally-Extended Exploration. Klissarov and Machado. 2023

[3] Learning Abstract Options. Riemer et al. 2018

---

> ### Author Response · Authors · 2023-11-18
> **Response to Questions**
>
> Thank you for your feedback.
>
> ## Clarify the sentence “The entropy term $H_{\phi}(Z|s_0)$ encourages the goal space policy $\mu_{\phi}(s_0)$ to output larger goal spaces.”
>
> The output of the goal space policy includes the half-lengths of each dimension of the uniform goal space distribution.  The entropy term $H_{\phi}(Z|s_0)$ in the Goal-Conditioned Empowerment objective incentivizes the goal space policy to output larger halfwidths, which is what we meant by “larger goal spaces.”  The dimensionality of the goal space is fixed and we can clarify that in the paper.
>
> ## Why is a model of the transition dynamics needed?
>
> The primary need for the model is to generate unbiased (state, subgoal action, next state) tuples for training the above-base level goal-conditioned policies with RL.  Given that the goal-conditioned policies at each level are continually changing, storing and sampling these transitions from a replay buffer would provide biased samples.
>
> In addition, we have tried to argue that existing empowerment-based skill-learning methods (e.g., DIAYN) that train the posterior distribution $q_{\phi}(z|s_0,s_N)$ with maximum likelihood will also need a model when the distribution of skill-start states $p(s_0)$ is highly entropic.  Maximum likelihood training of $q_{\phi}(z|s_0,s_N)$ requires on-policy tuples (s_0,z,s_N), which require executing the _current_ skill-conditioned policy $\pi_{\theta}(a|s,z)$ from the skill start state $s_0$ for $N$ actions.  Without a model, this would require executing the skill-conditioned policy for every skill $z$ at every start state $s_0$ to get unbiased (s_0,z,s_N) tuples.  This is not practical when p(s_0) and p(z|s_0) are highly entropic (i.e., when you need to learn a large set of skills from each state in a large set of states).  The alternative of storing the (s_0,z,s_N) tuples (e.g., p-HER [1]) in a replay buffer will provide biased samples because the skill-conditioned policy is continually changing.  Having a model would enable the user to execute the skill-conditioned policy from the necessary (s_0, z) combinations in parallel.
>
> ## Justify the statement "hand-crafted goal space may only achieve a loose lower bound on empowerment".
>
> We will break this down into two possible cases.  In the simpler case of the human trainer setting too small of a goal space (i.e., there are additional goals outside the hand-crafted goal space that can be achieved), $H(S_N|s_0)$ (the upper bound of empowerment assuming a discrete goal space) will be smaller than when the goal space includes all of the achievable goal states.  This would produce a loose lower bound on empowerment if the hand-crafted goal space is significantly smaller than the achievable goal space.
>
> In the case of the human trainer setting the goal space too large (i.e., there are goals in the goal space that cannot be achieved in $N$ steps), then the goals that cannot be achieved will yield skill end states $s_N$ that are redundant as they can be achieved by other goals (e.g., the goal $s_N$).  This will lower $H(S_N|s_0)$ unless all the redundant skill end states $s_N$ are distributed uniformly over the reachable state space $S_N$.  If the redundant states are not uniformly distributed across $S_N$, this again can produce a loose lower bound on empowerment.
>
> In our case of the continuous goal space distributions, it’s more complicated because there is no upper bound on the conditional entropy $H(S_N|s_0,Z)$, which can reach infinity.  But clearly in practice there will be situations when the above two cases apply and the empowerment that is achieved is significantly lower than what is possible with a learned goal space.
>
> ## Missing citation to previous work on HRL.
>
> Can you provide citation [4] again as the citation cannot be seen in your rebuttal?  Thanks.
>
> ## Results do not offer experiments that separate the effects of empowerment and the goal-conditioned hierarchical policies.
>
> We may be misinterpreting the question, but we are not sure how to separately assess them because we are not aware of other ways to build a goal-conditioned hierarchy in an unsupervised manner.  Our work shows that empowerment can be a way to generate each level of the goal-conditioned hierarchy in an unsupervised manner.
>
> [1] Choi et al. Variational Empowerment as Representation Learning for Goal-Based Reinforcement Learning. ICML 2021

---

> > ### Comment · Reviewer_2VaP · 2023-11-19
> >
> > Dear authors,
> >
> > the citation [4] was a typo and was actually referring to [3]. I have updated my original review to correct for this typo.
> >
> > I would encourage the authors to address all the points of my review in order to have a constructive conversation. As it stands many of my points in "Weaknesses" have been ignored and some of the "Questions" too.

---

### Official Review · Reviewer_CmJ8 · 2023-10-31

**Soundness:** 3 good
**Presentation:** 4 excellent
**Contribution:** 3 good
**Rating:** 6
**Confidence:** 3

**Summary:**

This paper considers the problem of maximizing empowerment for agents with a large set of skills which is challenging as the objective function is difficult to optimize. This paper firstly proposed a new variational lower bound objective for Goal-conditioned empowerment  that does not require handcrafted goal space in advance. Instead of a fixed goal space distribution,  the proposed new MI variational lower bound within goal-conditioned empowerment model the goal distribution with  parameterized uniform distribution.  Secondly, this paper proposes a hierarchical architecture for scaling the proposed Goal-conditioned empowerment to a longer time horizon. Through simulated robotic navigation tasks, the proposed framework outperforms the baselines.

**Strengths:**

The proposed hierarchical framework handle the long time horizon problem naturally with a well pre-designed hierarchical structure. By parameterizing the goal space distribution, the proposed method can learn the parameters for goal space distribution and goal-conditioned policy at the same time. The proposed method is valid and novel. This paper presents sufficient empirical evidence and simulations that validate the effectiveness of the proposed method.

**Weaknesses:**

1. The proposed hierarchical framework highly depends on designer's implementation which need strong expert knowledge. It won't be a scalable solution in general for long-time horizon RL tasks.
2. While the experiment setup can show the effectiveness of the proposed method, there remains a scope of further demonstration in more complex environments.

**Questions:**

The authors did not provide enough discussion on how to decide on k and how to design the structure. It will be great if the authors can provide explanations and more insight on this.

---

> ### Author Response · Authors · 2023-11-20
>
> Thank you for your feedback.
>
> We agree with your characterization of the strengths and weaknesses of the algorithm.  In regards to selecting the number of levels $k$, our main criteria was that we wanted each level to learn a short sequence of actions given the difficulties of credit assignment in RL.  Given that the $k$-th root of the number of actions in an episode, (i.e., $(\text{actions per episode})^{\frac{1}{k}}$), reflects the lengths of the policies each level needs to learn, we were looking for the $k$-th root to be some small number (ideally under 10).  Although increasing the number of levels can make learning long horizon skills more tractable, the main drawback is that it increases the wall clock time of training increases.  Training the top level goal-conditioned and goal-space policies in the hierarchy requires simulating $N^k$ primitive actions, which increases exponentially with $k$.   $N$ here refers to the maximum number of actions executed by each level.

---

### Official Review · Reviewer_zB9W · 2023-11-01

**Soundness:** 2 fair
**Presentation:** 3 good
**Contribution:** 3 good
**Rating:** 5
**Confidence:** 3

**Summary:**

This paper relates to the skill discovery problem of a reinforcement learning agent, which is an important topic in the domain. The authors follow the "Empowerment" approach, which maximizes the mutual information between skills and states, to enable an agent to acquire a diverse set of skills. The authors propose a new variational lower bound derived from the goal-conditioned policy formation, in order to make the mutual information optimization tractable. Successively, when expanding to long-horizontal tasks, the authors introduced a hierarchical approach.

Evaluations are done in a specially hand-modified simulation environment based on (Ant, Half Cheetah). The authors provide justifications for the selection of evaluation domains.

**Strengths:**

- Well-written backgrounds and motivations.
- Well-developed methodology: The reviewers find the methodology development of the paper easy to follow in Sections 2 and 3.
- The reduction of the second term in Eq.(2) to a goal-conditioned objective (Eq. (3) is novel to me and very easy to follow.

**Weaknesses:**

- The reviewer views the contribution of introducing goal-conditioned MDP formation as not novel, nor brings significant performance improvement, due to the two reasons.
(1) As a follow-up work of computing the Empowerment loss term, the key challenge is i) high-dimensional state-action space, and ii) without access to the environment's dynamic model. However, after examining the proposed method, the above two key challenges are not well addressed.
(2) instead, the authors propose a goal-conditioned policy formation, that makes the mutual information term an alternating optimization between goal-space policies and goal-conditioned policies, which seems reasonable to me, but the contribution is not significant.
- The reviewer found it hard to justify the effectiveness of the proposed method, due to the following issue.
(1) Only the Ant Field-small and Half-cheetah-small simulation tasks are evaluated against three baselines. If results on more complex tasks can be provided, the soundness of the method will be improved.

**Questions:**

Could the authors explain why the error bars in Fig. 3 and 4 have a large deviation?

---

> ### Author Response · Authors · 2023-11-20
>
> Thank you for your feedback.
>
> We agree with your characterization of the weaknesses of the algorithm.  The large standard deviations for the 1 skill level agent in the “Half Cheetah – Big” domain in Figure 3 and the 2 skill level agent in Figure 4 was due to there being a trial in which the agent failed to learn anything meaningful in phase 2.  The poor performance in phase 2 was a result of learning an inadequate skillset in phase 1 of training.  The large standard deviation for the 2 skill level agent in “Ant Field – Big” early in training is something we occasionally noticed in our experiments and is likely due to the difficulty of the phase 2 task.  In the large domains, the phase 2 task is a very sparse reward goal-achieving task.  Sometimes due to the lack of reward signal, the phase 2 policy will saturate the tanh activation function and then become stuck proposing subgoals at the extremes (in this case, the corners) of the learned goal space box.  The policy can become stuck because the gradients are near zero when the tanh activation function saturates.  Ultimately though, in all our experiments, the phase 2 policy of the agent with more levels policy always recovered.  A simple way to alleviate this issue would be to increase the goal-achieving threshold for the phase 2 task in order to make the reward signal less sparse.

---

### Meta-Review · Area_Chair_WD4b · 2023-11-30

**Metareview:**

**Summary**: A skill learning method based on mutual information that makes use of a hierarchy to solve longer horizon goal reaching tasks. The method involves a novel lower bound on mutual information. It is evaluated on simulated robotic tasks, where it outperforms baselines.

**Strengths**: The reviewers found the paper easy to read and acknowledged strong results on the tasks shown. They appreciated the novelty of the method, and that the method went beyond the standard 2-layer hierarchy.

**Weaknesses**:
The reviewers encouraged the authors to add additional baselines (esp. ones based on goal-conditioned RL) and evaluated on another complex task. They felt that the method required some ``expert knowledge'' and access to the environment dynamics. They provided another a number of writing suggestions.

**Justification For Why Not Higher Score:**

Most of the reviewers voted for rejecting the paper (3/5/5/6). It seems like the main contribution is empirical (the connection between GCRL and empowerment was noted in [Choi '21]), and the reviewers felt that additional baselines needed to be added their to make strong claims about the proposed method.

**Justification For Why Not Lower Score:**

N/A

---

### Decision · Program_Chairs · 2024-01-16

Reject